# Clinical Implications of High-Sensitivity Troponin Elevation Levels in Non-ST-Segment Elevation Myocardial Infarction Patients: Beyond Diagnostics

**DOI:** 10.3390/diagnostics14090893

**Published:** 2024-04-25

**Authors:** Constanza Bravo, Geovanna Vizcarra, Antonia Sánchez, Francisca Cárdenas, Juan Pablo Canales, Héctor Ugalde, Alfredo Parra-Lucares

**Affiliations:** 1Cardiovascular Department, Hospital Clínico Universidad de Chile, Santiago 8380456, Chile; 2Faculty of Medicine, School of Medicine, Universidad de Chile, Santiago 8380456, Chile; 3Cardiovascular Research Unit, Hospital Clínico Universidad de Chile, Santiago 8380456, Chile

**Keywords:** troponin, NSTEMI, coronary artery disease, obesity, acute heart failure

## Abstract

Standard troponin has long been pivotal in diagnosing coronary syndrome, especially Non-ST-Segment Elevation Myocardial Infarction (NSTEMI). The recent introduction of high-sensitivity troponin (hs-cTnI) has elevated it to the gold standard. Yet, its nuanced role in predicting angiographic lesions and clinical outcomes, notably in specific populations like obesity, remains underexplored. **Aim**: To evaluate the association between hs-cTnI magnitude in NSTEMI patients and angiographic findings, progression to acute heart failure, and its performance in obesity. **Methods**: Retrospective study of 208 NSTEMI patients at a large university center (2020–2023). Hs-cTnI values were assessed for angiographic severity, acute heart failure, and characteristics in the obese population. Data collected and diagnostic performance were evaluated using manufacturer-specified cutoffs. **Results**: 97.12% of patients had a single culprit vessel. Hs-cTnI elevation correlated with angiographic stenosis severity. Performance for detecting severe coronary disease was low, with no improvement using a higher cutoff. No association was found between hs-cTnI and the culprit vessel location. Hs-cTnI did not predict acute heart failure progression. In the obese population, hs-cTnI levels were higher, but acute heart failure occurred less frequently than in non-obese counterparts. **Conclusions**: In NSTEMI, hs-cTnI elevation is associated with significant stenosis, but not with location or acute heart failure. Obesity correlates with higher hs-cTnI levels but a reduced risk of acute heart failure during NSTEMI.

## 1. Introduction

For years, the cornerstone of diagnosing coronary syndrome has been the standard troponin [1]. Of these acute coronary events, 70% correspond to Non-ST-Segment Elevation Myocardial Infarction (NSTEMI) [2], a scenario where the role of this biomarker is crucial. Over time, various studies have shown an association between the magnitude of its elevation in an initial assessment in the emergency department and different clinical characteristics in these patients, such as the angiographic severity of the lesion in the culprit coronary vessel, its anatomical complexity, or the presence of intracoronary thrombus [3,4,5,6]. Previously, biomarkers that are not currently used routinely in acute coronary syndrome, such as CK-MB, were also correlated with the extent of infarction, echocardiographic features following the acute episode, and even mortality [7]. This correlation was likely associated with the macrostructural involvement that these biomarkers could determine. Nowadays, this seems impractical given the ongoing search for molecules that can identify myocardial damage at a microstructural level, allowing for better anticipation and prevention of ultrastructural damage [8].

Since the advent of high-sensitivity troponin (particularly, High-Sensitivity cardiac troponin I or hs-cTnI), this biomarker has evolved into the gold standard for diagnosing acute coronary pathology, particularly in NSTEMI patients, consistently retaining this designation across various updates in clinical guidelines from diverse cardiological societies [9,10]. Several studies have attempted to illustrate its nuanced variations in specific scenarios [11,12]. However, to our knowledge, none have systematically assessed whether the extent of high-sensitivity troponin elevation in the initial measurement, when patients present at the emergency department and are diagnosed with NSTEMI, can effectively forecast the characteristics of the angiographic lesion, or what these features are in special populations. This includes comparisons with other cardiac biomarkers such as amino-terminal pro-B-type natriuretic peptide (NT-proBNP), where levels in the obese population are comparatively diminished, especially in the context of acutely decompensated heart failure [13].

Despite the theoretical limitations in predicting the location of the culprit coronary lesion in NSTEMI through electrocardiographic changes, empirical findings indicate that up to 25% of diagnosed patients exhibit vessel occlusion, a pattern like that observed in ST-Segment Elevation Myocardial Infarction (STEMI) [14,15]. Furthermore, a frequent practice in emergency departments involves assigning severity to NSTEMI based on initial measurements of hs-cTnI that exceed the upper normal limit, often lacking a robust rationale for such classification (the higher, the more severe). These observations underscore the need for further investigation and critical evaluation of current clinical practices to enhance diagnostic precision and patient care in the context of NSTEMI.

Considering previous data, this study aimed to evaluate the association between the magnitude of high-sensitivity troponin in patients with NSTEMI and angiographic findings in coronary arteries indicative of severe disease. Furthermore, we sought to evaluate their risk of progression to various stages of acute heart failure during hospitalization, as classified by the Killip-Kimball classification, and their behavior within the obese population. This investigation was conducted within a cohort of patients diagnosed with NSTEMI at a university hospital, providing valuable insights into the scientific understanding of these clinical parameters.

## 2. Materials and Methods

### 2.1. Study Design

This was a retrospective observational study in a large university center of 1401 medical records of patients admitted to the catheterization laboratory between January 2020 and July 2023. Demographic, clinical, laboratory, and angiographic variables were collected and managed using RedCap electronic data capture tools hosted at the Facultad de Medicina of the Universidad de Chile [16,17]. The cutoff values of hs-cTnI VITROS^®^ (Ortho Clinical Diagnostics, Rarity, NJ, USA) used to assess the performance of this biomarker in determining angiographic findings of significant coronary stenosis were based on the usual cutoff points recommended by the manufacturer, as employed in the diagnosis of NSTEMI (p99 of hs-cTnI or 11 ng/L, and 5 times the value to enhance the PPV or 50 ng/L) [18,19]. This study received approval from the institution’s ethics committee for accessing information from patients’ medical records (CEIC HCUCh—55/2023).

### 2.2. Inclusion and Exclusion Criteria

The study included all patients who were discharged from the hospital who were 18 years of age or older and were diagnosed with NSTEMI. The diagnosis of NSTEMI was based on a combination of clinical presentation, ECG findings, elevated hs-cTnI levels, and coronary angiographic study as a gold standard. These diagnostic criteria were all thoroughly assessed to include the most accurate and relevant patient population in the study.

Patients who were studied in the catheterization laboratory for procedures other than coronary angiography (right cardiac catheterization, prosthetic valve implantation, left atrial appendage closure, etc.) were excluded from the study. Additionally, those with conditions that could alter the baseline value of hs-cTnI were excluded such as end-stage chronic kidney disease, myocarditis, pericarditis, post-cardiac arrest, infiltrative disease, arrhythmia, sepsis, acute bleeding, endocrine disorders, or others according investigators criteria.

### 2.3. Statistical Analysis

For the statistical analysis discrete variables were expressed as absolute values (percentages), and continuous variables were expressed as arithmetic mean ± standard deviation or median [p25–p75]. For comparison data between groups, the chi-square test for discrete variables was used, or the *t*-test for unpaired groups. To assess potential confounders, a multivariate linear regression analysis was conducted. To evaluate the performance of hs-cTnI in the angiographic finding of significant stenosis, a ROC curve was constructed, and sensitivity, specificity, positive predictive value (PPV), negative predictive value (NPV), and accuracy were determined. Data analysis was performed using GraphPad Prism v.10.0 (GraphPad Software, La Jolla, CA, USA) and Stata/MP v.14.1 (Stata Software, College Station, TX, USA). All analyses were two-tailed, and a statistically significant difference was defined as a *p*-value below 5% (*p* < 0.05).

## 3. Results

### 3.1. Study Population and Baseline Characteristics

A review of 1401 medical records of patients who underwent procedures in the catheterization laboratory of our university center between January 2020 and July 2023 was conducted. From the total reviewed medical records, 208 patients meeting the pre-established inclusion criteria were selected, as indicated in Figure 1. Subsequently, the baseline characteristics of the patients included in the study were summarized, as shown in Table 1.

The group of individuals included in the study had an average age of 69.5 years, with a majority being males (74.52%). The cohort presented with common comorbidities such as hypertension (66.83%), diabetes (45.19%), and smoking (32.19%). Upon laboratory analysis, the mean estimated glomerular filtration rate was found to be 57 [mL/min], and serum creatinine was 0.98 [mg/dL]. The hs-cTnI median stood at 122.75 (IQR: 31.17–777.9) [ng/L]. Clinical findings indicated a high prevalence of severe angiographic outcomes (87.02%) and diverse involvement of culprit coronary arteries, with the left anterior descending artery being the most frequently affected (54.46%). As per the report, the majority of patients did not show progression in stages of acute heart failure related to the specific coronary event at the time of admission to the hospital unit, as determined by the Killip-Kimbal classification.

### 3.2. Angiographic Severity of Coronary Lesion in the Culprit Artery

Out of the 208 analyzed patients, 202 were attributed to a single vessel as the culprit of the acute coronary event (97.12%).

Regarding the association between angiographic severity and the magnitude of hs-cTnI elevation, patients were grouped into none to moderate lesion (coronary vessel stenosis < 70%, *n* = 27), severe lesion (culprit vessel stenosis 70–89%, *n* = 112), and critical/occluded vessel (stenosis 90–100%, *n* = 69). This grouping was based on the angiographically observed compromise in coronary flow and its clinical repercussions. It was observed that the magnitude of hs-cTnI elevation in the initial emergency measurement was associated with the degree of stenosis observed in coronary angiography, with a greater involvement as the magnitude increased among groups (*p* = 0.021, F 3.933). The average elevation magnitude for each group can be seen in Figure 2A.

### 3.3. Validation of hs-cTnI for the Identification of Angiographically Severe Lesions

Based on the previous findings, we decided to evaluate hs-cTnI in terms of its performance in detecting angiographically severe coronary disease in coronary angiography using the same cutoff point defined by the manufacturer as the upper normal limit in diagnostic terms. Angiographically severe lesions were defined as coronary stenosis > 70% in any vessel (RCA, LCx, or LAD) or left main coronary artery lesion with >50% stenosis as reported in the catheterization laboratory. Similarly, we assessed the cutoff point set at five times above the upper normal limit since, for diagnostic purposes, it enhances its performance in terms of positive predictive value. Therefore, we hypothesized that it could also improve its performance in detecting angiographically severe disease. Unfortunately, the performance of hs-cTnI in this scenario is low (ROC AUC = 0.98, 95% CI 0.449–0.670) (Figure 2B), and using the cutoff point of 50 ng/L does not significantly enhance its performance, even reducing its precision (Table 2).

### 3.4. Localization of Culprit Artery Territory and Heart Failure at Admission

Subsequently, we hypothesized that the magnitude of hs-cTnI elevation might be associated with the percentage of compromised myocardial mass, and the latter could be related to the artery that irrigates that territory (given that 175 patients presented with a lesion of at least 70% in angiography, with cause attributed to the diseased vessel). Therefore, the coronary vessel irrigating a greater amount of myocardial mass could generate a higher biomarker elevation magnitude, and vice versa. Traditionally, territories have been defined as those irrigated by LAD (*n* = 109, 56.4%), and the rest (LCx and RCA) as non-LAD (*n* = 84, 43.5%), due to the lower percentage of myocardial mass they perfuse in the heart. Additionally, the analysis included the group of patients with LMCA involvement, as few data were observed regarding the total sample (*n* = 8, 4% of the total) and involving a broad territory. Therefore, they were added to the LAD-irrigated group. When comparing means in these two groups, no differences were found regarding the magnitude of hs-cTnI and the culprit vessel (*p* = 0.239, with LMCA, Figure 3A; *p* = 0.2595, without LMCA, Figure 3B).

We decided to determine whether the magnitude of hs-cTnI at the first measurement upon admission to the emergency department predicted the progression to acute heart failure in NSTEMI patients according to the Killip-Kimbal classification. It is anticipated that a higher level of the biomarker could represent a greater amount of compromised myocardium and, therefore, impairment of systolic function. Out of the total patients, only 37 presented Killip-Kimbal > 1 (17.79%), and considering the clinical significance (symptoms associated with Killip-Kimbal > 1), we decided to group patients at Killip-Kimbal = 1 or >1 for analysis. A *t*-test was conducted for the means of the described groups, revealing no differences between them (log_10_ hs-cTnI [ng/L] 2.243 vs. 2.186, *p* = 0.743) (Figure 3C).

### 3.5. Performance of hs-cTnI and Characterization of Clinical Outcomes in the Obese Population

Notably, the performance of a biomarker within specific populations gains significance, particularly considering previously reported modifications observed in individuals with obesity (BMI ≥ 30 kg/m^2^). Based on the obtained data, we evaluated the characteristics of hs-cTnI according to the presence of obesity in NSTEMI patients. In our sample, 29.3% were obese with an average BMI of 33.79 ± 3.72 kg/m^2^. Analyzing the means of these groups, differences were found in the magnitude of hs-cTnI elevation in the first measurement at the emergency room, with higher values for the obese patient group (log_10_[hs-cTnI] 2.14 vs. 2.45; *p* = 0.0287, 95% CI 0.03295–0.5949 ng/L), as shown in Figure 4A. After adjusting for confounding variables of hs-cTnI and obesity about other analyzed variables, this trend persists (Table 3).

Similarly, we decided to investigate the severity of the lesions found when there is a culprit artery for the coronary ischemic condition in this population, compared to their non-obese counterparts using the same positive coronary angiography classification as described previously, finding no differences (*p* = 0.6531) (Figure 4B).

On the other hand, we explored the frequency of progression events to heart failure according to the Killip-Kimbal classification = 1 or >1. We observed a higher frequency of Killip-Kimbal > 1 events in non-obese patients (*p* = 0.0271) as shown in Figure 4C.

## 4. Discussion

To our knowledge, only a few studies have systematically evaluated the characteristics of hs-cTnI beyond its diagnostic capabilities and out of the NSTEMI scenario [20,21,22,23,24,25]. It is worth mentioning that hs-cTnI is a more sensitive test than the standard troponin test, allowing for earlier detection of myocardial injury. This has led to a decrease in the time required for diagnosis and an increase in the accuracy of diagnosing NSTEMI, successfully reducing the prevalence of unstable angina, as discussed in the literature [26].

In this study, a retrospective analysis of 208 NSTEMI patients in a large university center established the predictive capacity of hs-cTnI concerning angiographic findings of coronary stenosis > 70% in the culprit vessel. Notably, over 95% of patients in this sample had one of the coronary vessels responsible for NSTEMI, emphasizing the challenge of discriminating the location of the culprit vessel with subendocardial ischemia [27]. This data is very interesting because it is similar to other reports where achieving identification of a culprit artery through angiography study is possible in more than 70% of cases analyzed [28,29], including intracoronary imaging strategies [30,31]. This contrasts with what was previously proposed in the literature, probably due to lower efficiency in coronary angiographic study techniques [32]. Moreover, within the general population characteristics, there was a low percentage of medication use recommended for high-risk individuals, such as those with coronary heart disease or diabetes. Statins, for instance, were used in less than half of these patients, similar to other reports [33,34].

An important finding was the correlation between the magnitude of the initial hs-cTnI measurement in the emergency department and the identification of a critical coronary stenosis or occluded vessel in angiographic studies. This correlation was particularly evident when hs-cTnI values were high, an observation in previous reports [18]. However, when we analyzed the performance of hs-cTnI in predicting angiographically severe stenosis or greater involvement using a cutoff similar to the diagnostic cutoff value, its accuracy was low. Attempting to enhance its positive predictive value (PPV) by using a value five times above the p99 (consistent with diagnostic recommendations) [19] only resulted in decreased precision without improvement.

It has been discussed that the elevation of the biomarker is related to the amount of myocardial tissue compromised, and in turn, with the coronary vessel that irrigates it. This is because it has been estimated that these arteries differentially supply blood to the myocardium, with almost half of it supplied by the anterior descending artery (LAD) [35]. Therefore, it could be possible to identify and categorize myocardial tissue into LAD and non-LAD territories. However, our study suggested that the location of coronary stenosis is not associated with the biomarker level, possibly due to the variation between patients in the myocardial mass irrigated by each coronary artery or the lack of precision in coronary artery obstruction details in our records. There is no impact when we include data related to LMCA, because there are few patients with involvement in this area.

Concerning clinical outcomes, earlier data indicated that patients admitted for acute coronary syndrome developed heart failure according to elevated hs-cTnI levels [36], likely influenced by the inclusion of STEMI patients. In our study involving NSTEMI patients alone, no difference was observed in the Killip-Kimbal classification stages between the two groups (Killip-Kimbal = 1 or >1). Consequently, our findings suggest that the initial hs-cTnI measurement did not predict an adverse clinical outcome related to the development of acute heart failure during the admission of these patients. This contrasts with standard troponin [29], which includes mortality associated with higher levels, primarily due to the greater structural damage linked to the elevation of standard troponins in the era before the ultrasensitive ones. This observation aligns with analogous findings in patients with myocardial infarction with non-obstructive coronary arteries (MINOCA) [37], marking it as one of the most feared complications in this patient setting.

Some previous reports with other biomarkers (including standard and high-sensitive troponins) suggest that there may be differences, particularly in an important population for acute coronary pathology, such as obese individuals [13,38]. When analyzing the performance of hs-cTnI in this population, our data revealed an upward difference in this biomarker compared to their non-obese counterparts. This finding is consistent with previous studies on hs-cTnT [32], in which this phenomenon could be related to the remodeling process in the heart that continues over time. Nevertheless, this study did not find a difference in the progression of Killip-Kimbal stages between both groups in the acute coronary setting. It is noteworthy that some studies indicate that obesity might even play a protective role in the development of acute heart failure, a phenomenon still not clearly explained [39,40].

This study has several limitations, including focusing on a single initial hs-cTnI measurement and irregular control after angiographic studies. The above is a challenging limitation to avoid, as a percutaneous resolution of reported coronary lesions generates an early peak in hs-cTnI values and can induce a misinterpretation of the rising levels; for this reason, it is not routine for us to perform a biomarker follow-up after a coronary angioplasty procedure. Similarly, not all centers have catheterization equipment to address this pathology, making the applicability of our work useful for low-complexity centers and the possible courses of action to be taken in the face of a patient with this diagnosis, relying solely on the initial determination of hs-cTnI. Additionally, with the advent of ultrasensitive techniques, we have observed a decrease in the need for a second draw with the data from patients at our center, as they allow for immediate categorization as either rule in or rule out. Additionally, retrospective studies can have an inherent bias due to issues with data acquisition, such as a lack of information about the time between when symptoms started and when the patient sought emergency care. This means that patients may have initially gone to other healthcare facilities with different complaints, or they may not remember the exact timing of events leading up to their visit to our hospital. This can create a reporting bias in terms of the symptoms they experienced. This issue has been documented in various studies related to this medical context. Finally, previous articles have reported an association between the peak of standard troponin levels and 3-vessel disease in NSTEMI patients, an aspect that we cannot evaluate in this first part of our work due to its initial design.

## 5. Conclusions

In conclusion, our study suggests that, in NSTEMI patients, the magnitude of hs-cTnI elevation in the first measurement may be directly associated with significant stenosis in the culprit vessel (>90%). However, it may not be associated with the location of this lesion or its clinical impact on acute heart failure development. In obese individuals, hs-cTnI levels appear higher than in the non-obese population. Nevertheless, they less frequently develop acute heart failure during an NSTEMI than those without this condition. In any case, it is necessary to generate population databases that allow for complementing the findings obtained to support the observed results.

## Figures and Tables

**Figure 1 diagnostics-14-00893-f001:**
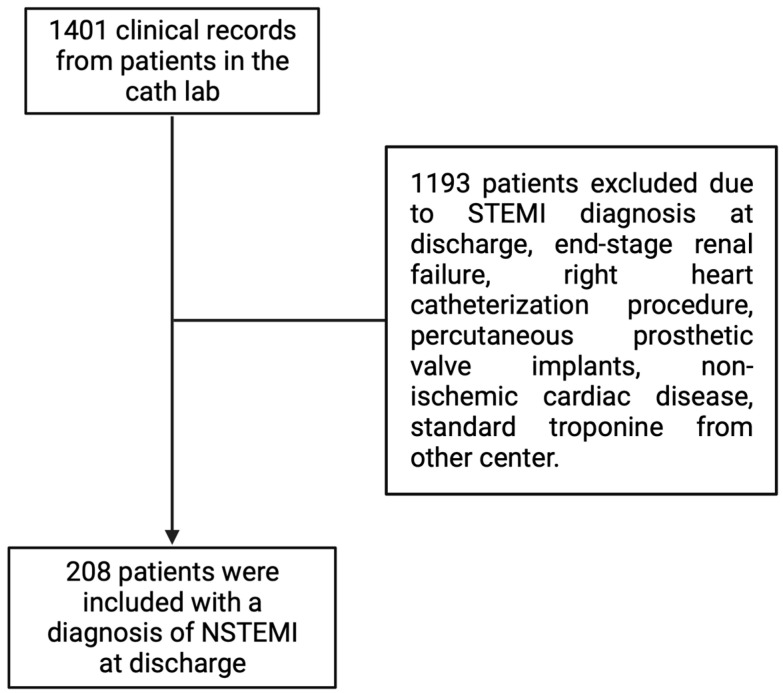
Study design.

**Figure 2 diagnostics-14-00893-f002:**
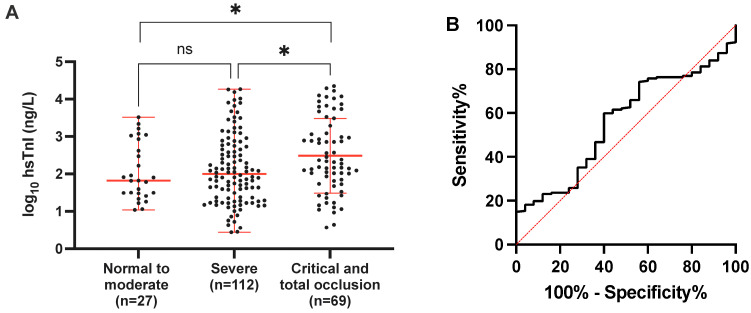
The magnitude of hs-cTnI elevation levels and severity of coronary angiography stenosis. In panel (**A**), patients were classified based on the severity of coronary stenosis by coronary angiography into normal to moderate (0–69%), severe (70–89%), and critical to total occlusion (90–100%) categories according to the clinical impact of the finding. The mean ± SD of log_10_[hs-cTnI] (ng/L) is presented, and the means of the value for each group were compared using ANOVA with Tukey’s post hoc test. In panel (**B**), a ROC curve analysis is shown for the performance of hs-cTnI in detecting significant stenosis (defined as any coronary artery with stenosis > 70% and LMCA > 50%). Ns = not significant, * *p* < 0.05.

**Figure 3 diagnostics-14-00893-f003:**
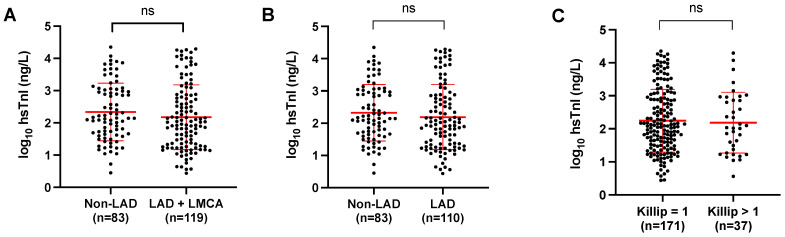
The magnitude of hs-cTnI elevation levels: highlighting the localization of the culprit coronary artery and the development of acute heart failure. In panels (**A**,**B**), patients were categorized based on the culprit coronary artery into LAD and non-LAD groups. In panel (**A**), we included LMCA in the LAD group due to its involvement in a large territory, but it was excluded in panel (**B**) due to a limited amount of data. Panel (**C**) compares troponin levels concerning acute heart failure related to NSTEMI during presentation in the emergency department. The patients are grouped into Killip-Kimball 1 or >1 based on clinical relevance. The mean ± SD of log_10_[hs-cTnI] (ng/L) is presented, and the means of the value for each group were compared using *t*-test. Ns = not significant.

**Figure 4 diagnostics-14-00893-f004:**
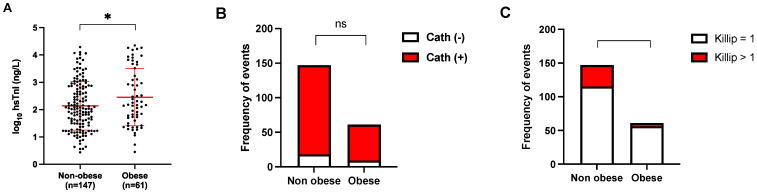
Obesity, hs-cTnI pattern, and clinical outcomes. Figure (**A**) illustrates first troponin levels in the context of NSTEMI in the obese population compared to the non-obese counterpart at presentation in the emergency department. In (**B**), the differences in the frequency of significant coronary angiographic finding (red) are shown in the obese versus non-obese population. Finally, in (**C**), the frequency of events related to the progression to acute heart failure (Killip-Kimbal = 1 or Killip-Kimbal > 1) is depicted about NSTEMI in the same two populations described. In (**A**), the mean ± SD of log_10_[hs-cTnI] (ng/L) is presented, and the mean values for each group were compared using *t*-test. In the case of (**B**,**C**), contingency tables were analyzed using Fisher’s exact test. * *p* < 0.05, ns = not significant.

**Table 1 diagnostics-14-00893-t001:** Characteristics of recruited patients at baseline.

Characteristics of Patients at Baseline	
Demographic	
Age—years ± SD	69.5 ± 11.47
Female—*n* (%)	53 (25.48)
Comorbidities	
Hypertension—*n* (%)	139 (66.83)
Diabetes—*n* (%)	94 (45.19)
Smoking—*n* (%)	68 (32.19)
Obesity—*n* (%)	61 (29.33)
Coronary artery disease—*n* (%)	68 (32.69)
Statins use—*n* (%)	82 (39.42)
Laboratory	
eGFR—mean ± SD [mL/min]	57 ± 8.39
Serum creatinine—mean ± SD [mg/dL]	0.98 ± 0.5
hs-TnI—median (IQR) [ng/L]	122.75 (31.17–777.9)
Clinical	
Severe in angiography	
Positive—*n* (%)	181 (87.02)
Killip-Kimbal classification	
Killip-Kimbal 1—*n* (%)	171 (82.21)
Killip-Kimbal 2—*n* (%)	21 (10.09)
Killip-Kimbal 3—*n* (%)	14 (6.73)
Killip-Kimbal 4—*n* (%)	2 (0.96)
Culprit coronary artery	
RCA—*n* (%)	51 (25.25)
LAD—*n* (%)	110 (54.46)
LCx—*n* (%)	32 (15.84)
LMCA—*n* (%)	9 (4.46)

**Table 2 diagnostics-14-00893-t002:** Calculated diagnostic performance of hs-cTnI with cutoff points of 11 and 50 (ng/L). The data were obtained from ROC curve analysis, and the selected cutoff values were the same as those used in the diagnostic criteria for NSTEMI (p99 for 11 ng/L [18]) and recommended in NSTEMI guidelines to improve PPV in diagnosis [19].

Cut Off	Sensibility (%)	Specificity (%)	PPV (%)	NPV (%)	Precision (%)
11 (ng/L)	93.37%	3.7%	86.67%	7.69%	81.73%
50 (ng/L)	70.17%	44.44%	89.44%	18.18%	66.83%

**Table 3 diagnostics-14-00893-t003:** Multivariate analysis of covariates associated with hs-cTnI and obesity. Several potential demographic and clinical variables are associated with the variation in the magnitude of hs-cTnI and obesity. The body mass variable changes linearly concerning the hs-cTnI variable without losing its statistical significance when analyzed with the rest of the covariates. The coefficient, the 95% confidence interval (95% CI), and the *p*-value are detailed in the table. * *p* < 0.05.

	Multivariate Analysis			
Variable	Coef.	IC 95%		*p*-Value
Age	−44.52	−97.948	8.908	0.102
Gender	508.62	−717.683	1734.929	0.414
Type 2 Diabetes Mellitus	166.96	−934.275	1268.203	0.765
Arterial hypertension	419.01	−806.376	1644.399	0.501
Smoking	495.15	−691.289	1681.598	0.412
Obesity	1638.36	438.994	2837.73	0.008 *
Estimated glomerular filtration rate	−9.71	−73.096	53.672	0.763

## Data Availability

The information that supports the findings of our study is available upon reasonable request to the corresponding author, A.P.-L. This information is not publicly available because it contains data that may compromise the privacy of the research participants and third-party restrictions.

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
