# Peer review of "Clinical Implications of High-Sensitivity Troponin Elevation Levels in Non-ST-Segment Elevation Myocardial Infarction Patients: Beyond Diagnostics"

_diagnostics, 2024, doi:10.3390/diagnostics14090893_

Round 1
Reviewer 1 Report
Comments and Suggestions for Authors
I have reviewed the manuscript titled "Clinical Implications of High-Sensitivity Troponin Elevation Levels in NSTEMI Patients: Beyond Diagnostics" and appreciate its well-structured presentation on this important subject. However, I have several concerns that I believe need addressing to enhance the quality of the manuscript. These concerns are outlined below:
Population Representation: The manuscript discusses NSTEMI patients, yet only 14% of the patients had NSTEMI during the study period. It would be valuable for the authors to clarify why NSTEMI patients were underrepresented in the study, particularly with cath lab volume. The literature reports that the most common clinical presentation for acute coronary syndrome is the NSTEMI.
Cardiogenic Shock: While cardiogenic shock was listed as an exclusion criterion, Table 1 indicates data on Killip-Kimbal IV, which signifies cardiogenic shock. This discrepancy requires clarification.
Coronary Involvement: The manuscript states that around 97% of patients had single-vessel disease, which differs from the typical presentation of NSTEMI, where multivessel involvement is more common. The authors should provide insights into why their population deviates from this norm.
Troponin Significance: Elevated troponin levels in ACS indicate myocardial injury, but their correlation with coronary involvement is challenging. While troponin levels have been shown to correlate with 3-vessel disease in NTEMI patients in previous studies (Correlation of cardiac troponin I levels (10 folds upper limit of normal) and extent of coronary artery disease in non-ST elevation myocardial infarction. J Pak Med Assoc 2010 Jun;60(6):423-8), the lack of correlation in the current study may be due to the predominance of single-vessel disease.
Novelty and Population Influence: The manuscript's contribution to the field is limited. The potential influence of the patient population on the observed results should be considered a potential bias.
Comments on the Quality of English Language
I recommend thoroughly reviewing the manuscript for spelling and grammar errors to ensure Clarity and readability.
Author Response
See in pdf file

Reviewer 2 Report
Comments and Suggestions for Authors
Thank you for possibility to review for Diagnostics.
The manuscript by Bravo et al. is written in a proper style with interesting analysis of performed study. The study was well designed, though I have doubt on the chosen timing of troponin examination, as the maximal one would be even more beneficial. However, the authors also found this issue as important and precisely described it in the limitation section.
The introduction is enriched with proper citations. The discussion section begins with some repetitions from the Introduction - it might be considered to re-write.
I have some questions and comments:
Why only patients who were discharged were included? did you include patients who died in course on NSTEMI during hospitalization
Table 1. Please correct Laboratorio to English version
ONLY statins are described in the table, so it is confusing for the reader to evaluate authors' statement in the discussion section regarding use of pharmacotherapy.
Did the authors tried to compare patients with KK class <1 to those >=3?
Generally, these minor comments do not undermine the paper value, as the analysis is properly performed and clearly described.
Author Response
See in PDF file

Reviewer 3 Report
Comments and Suggestions for Authors
The manuscript titled "Clinical implications of elevated levels of high-sensitivity troponin in NSTEMI patients: beyond diagnostics" describes the relationship between the magnitude of high-sensitivity troponin (hs-cTnI) in patients with non-ST-segment elevation myocardial infarction (NSTEMI), angiographic findings, the development of acute heart failure, and its effectiveness in obesity with a large patient sample. They noted that hs-cTnI did not predict the course of acute cardiac failure and that there was no correlation between it and the location of the culprit vessel. Although hs-cTnI levels were higher in the obese group than in their non-obese counterparts but acute heart failure occurred less frequently than in non-obese counter parts. In this study the author conducted a retrospective analysis of 208 NSTEMI patients at a large university center facility between 2020 and 2023. Hs-cTnI values were evaluated in relation to acute heart failure, angiographic severity, and features within the obese population. Overall, the study is well-structured, methods and data presented appear to be robust and consistent. The publications offer fresh information on the correlation between elevated high-sensitivity troponin and various coronary syndromes. The articles provide new insight regarding association with high-sensitivity troponin elevation in different coronary syndrome.
Minor question
cTnI and cTnT are unique to the heart, despite the fact that the heart contains three different isoforms of troponin. Because of this characteristic, troponin testing is now considered the gold standard for determining ruling-in or ruling-out patients with suspected acute coronary syndrome in the Emergency Department.Thus, I would like to know why the authors decided to use high-sensitivity troponin (hs-cTnI) for the investigation rather than the other isoform, cTnT.
Author Response
Response in PDF file

Round 2
Reviewer 1 Report
Comments and Suggestions for Authors
I want to thank the authors for their responses to my questions.